# Integrated genomic analysis reveals aberrations in WNT signaling in germ cell tumors of childhood and adolescence

Lin Xu [1,2,3,23] ✉, Joshua L. Pierce [3,23], Angelica Sanchez[3,23], Kenneth S. Chen [3], Abhay A. Shukla[3], Nicholas J. Fustino[3,22], Sarai H. Stuart [3], Aditya Bagrodia[4,5], Xue Xiao[1,2], Lei Guo[1,2], Mark D. Krailo[6,7], Furqan Shaikh[8], Deborah F. Billmire[9], Farzana Pashankar [10], Jessica Bestrashniy[11], J. Wolter Oosterhuis [12], Ad J. M. Gillis[12], Yang Xie[1,2,13], Lisa Teot[14], Jaume Mora [15], Jenny N. Poynter[11], Dinesh Rakheja [3,16], Leendert H. J. Looijenga[12], Bruce W. Draper[17], A. Lindsay Frazier [18] & James F. Amatruda [19,20,21] ✉

Germ cell tumors (GCTs) are neoplasms of the testis, ovary and extragonadal sites that occur in infants, children, adolescents and adults. Post-pubertal (type II) malignant GCTs may present as seminoma, non-seminoma or mixed histologies. In contrast, pre-pubertal (type I) GCTs are limited to (benign) teratoma and (malignant) yolk sac tumor (YST). Epidemiologic and molecular data have shown that pre- and post-pubertal GCTs arise by distinct mechanisms. Dedicated studies of the genomic landscape of type I and II GCT in children and adolescents are lacking. Here we present an integrated genomic analysis of extracranial GCTs across the age spectrum from 0–24 years. Activation of the WNT pathway by somatic mutation, copy-number alteration, and differential promoter methylation is a prominent feature of GCTs in children, adolescents and young adults, and is associated with poor clinical outcomes. Significantly, we find that small molecule WNT inhibitors can suppress GCT cells both in vitro and in vivo. These results highlight the importance of WNT pathway signaling in GCTs across all ages and provide a foundation for future efforts to develop targeted therapies for these cancers.

Germ cell tumors (GCTs) are neoplasms of the testis, ovary, and extragonadal sites that occur in infants, children, and adults[1]. GCTs are thought to originate from primordial germ cells (PGCs), which are cells with retained pluripotency that can be reprogrammed to embryonal stem cells with different developmental capacities[2]. GCTs retaining features of PGCs are known as seminomas ('dys-germinomas' in the ovary and 'germinomas' in extragonadal sites). In contrast, non-seminomatous GCTs are composed of embryonal carcinoma (EC), the stem cells that can differentiate in various differentiation lineages, including teratoma, yolk sac tumor (YST), and

choriocarcinoma (CC). Mixed malignant GCTs (MMGCTs) contain more than one histologic subtype and may be combined with a seminoma component. GCT have been divided into two major and clinically relevant subtypes: pre-pubertal (type I), and post-pubertal (type II), which also segregate by age at presentation, cytogenetic abnormalities, and histologic subtype[2]. Type I tumors are limited to teratomas and yolk sac tumors, while Type II tumors may contain seminoma and non-seminoma subtypes. One model that takes these differences into account posits that the type I GCT recapitulates features of primed embryonic stem cells (ESCs) with restricted

developmental potential, whereas type II GCT resembles the broader developmental potential of naïve ESC[3,4].

Epidemiologic and molecular data suggest that pediatric and adult GCTs may arise by distinct mechanisms or from different stages of PGC development[1]. Cytogenetic data consistently show loss of chromosomes 1p and 6q in Type I tumors, while Type II tumors commonly exhibit gain and sometimes regional amplification of the short arm of chromosome 12 (i.e., 12p)[5–7]. Regardless of these differences, malignant Type I and Type II GCTs are treated with the same cytotoxic chemotherapy regimens[1]. Platinum-based therapies have been very effective in the treatment of GCTs[8], but often cause severe side effects including hearing loss, kidney damage, and elevated risk of second malignancies[9,10]. In addition, cisplatin-based therapy is ultimately ineffective in up to 15% of patients[11]. However, no effective targeted molecular therapies have been approved for treating GCTs. Therefore, identifying therapeutic targets for GCTs is an urgent priority for improving outcomes for these patients.

Relatively few somatic mutations have been described in type II GCTs. The most commonly reported mutated gene is *KIT*, a tyrosine kinase growth factor receptor important for germ cell development[12,13]. Mutations have also been reported in *NRAS* and *KRAS*, signaling components of the MAP kinase pathway that act downstream of *KIT*[14–16]. Central nervous system Type II GCTs exhibit recurrent mutations in *KIT*, *RAS*, and *MTOR*[17]. A study of 42 adult testicular GCTs TGCTs revealed somatic mutations in *CDC27* and demonstrated that mutations in *XRCC2* are associated with cisplatin resistance[18]. Taylor-Weiner and co-workers identified recurrent chromosome arm-level amplifications and reciprocal loss of heterozygosity in testicular GCT (TGCT)[19], and a TCGA analysis of TGCT revealed mutations in *KIT*, *KRAS* and *NRAS*[16]. Recently, frequent gain of chromosome 3p25.3 has been described in cisplatin-resistant non-seminoma tumors[20]. These studies have overwhelmingly focused on adult men with testicular cancer, making it unclear to what extent Type 2 tumors of adolescents share the same molecular features. To date, no large-scale studies have described the mutational spectrum of extracranial type I GCT or ovarian GCTs. Here we report the genomic analysis of 145 primary GCTs of childhood and adolescence, including 70 type I GCTs and 75 type II GCTs of adolescents. For comparison, we also evaluated 64 type II GCTs from patients aged 19–24 years, and 20 ovarian YSTs (classified as Type II if the tumor exhibited chromosome 12p gain). We performed whole-exome sequencing on a set of tumors matched with normal tissue, complementing these results with a panel-based deep sequencing, copy-number analysis, methylation profiling and RNA-seq. Integrated analysis of these data revealed a pattern of somatic mutations, copy-number alterations, differential methylation and gene expression that together mediate increased activity of the WNT signaling pathway in GCTs. WNT activation appears to carry prognostic significance in GCTs of both children and adults, and thus may be a node for targeted therapies of these cancers.

## Results

To identify somatic mutations, we performed whole-exome sequencing on tumor-normal pairs of 50 GCT patients aged 0–18 years (discovery cohort, Supplementary Tables 1 and 2). We identified 1180 somatic mutations in total, including 299 somatic single-nucleotide variants (SNVs) and 19 somatic small-scale insertion/deletions (INDELs) that were predicted to be protein-altering. To validate mutations in the discovery cohort and study the prevalence of these mutations in a larger cohort, we performed custom-capture deep sequencing in 129 GCTs (48 out of 50 cases of the discovery cohort with sufficient DNA and an additional 81 GCTs) (Supplementary Table 1). Therefore, 131 GCT cases were studied by either whole exome or targeted deep sequencing. The 51 genes chosen for the validation set were selected because they were frequently mutated in the discovery cohort or are candidate GCT drivers based on previous studies[15,17,21,22]. The mutational landscape of these candidate driver genes is shown in Fig. 1a and Supplementary Data 1.

Type I, Type II and ovarian GCTs in our datasets had a low mutation rate (0.23 non-silent mutations per Mb on average), consistent with previous reports from adult type II TGCT[16,18,23,24]. All histologic subtypes have low mutation rates that are not statistically different from one another (Fig. 1b). The most common recurrently mutated gene was *KIT*, with nine mutations identified in seven patients. Similar to previous reports[13], all *KIT* mutations were identified in seminoma patients (Fig. 1a). Among these mutations, seven are in exon 17 (encoding the kinase activation loop), one is in exon 11 (encoding the regulatory domain of enzyme) and one is in exon 2 (encoding the Ig-like-C2-type 1 domain; Fig. 1c). In addition, *KRAS* mutations were found in two EC, four seminoma, two MMGCT, and two YST, and *NRAS* mutations in one seminoma and one YST. Consistent with previous reports[25], *KRAS*, *NRAS*, and *KIT* mutations were mutually exclusive within a given tumor.

The most striking finding was the prevalence of mutations in six WNT pathway genes (*CTNNB1, APC, LRP5, TCF7L2, CHD8*, and *FAT1*) in ten YSTs and three MMGCTs containing YST elements. Previous studies in other tumor types have suggested that loss-of-function mutations in *APC*[26], *CHD8*[27], and *FAT1*[28] could all activate the WNT pathway and promote tumorigenesis. We observed missense mutations in these six WNT genes, as well as stop-gain mutations in *APC* and *FAT1* that likely result in truncated proteins (Table S3). Moreover, we also found mutations of *FAT2, FAT3*, and *FAT4*, which share high sequence similarity to the *FAT1* gene and have been proposed as candidate tumor suppressors[29]. By integrating RNA-seq and whole-exome sequencing data, we found a significantly increased *CTNNB1* expression in GCT cases with somatic protein-altering mutations in FAT2 or FAT3 genes compared to GCT cases without such mutations (Mann–Whitney U test, $P = 0.001$; Supplementary Fig. 1). *FAT* family genes were mutated in Type I GCTs, whereas *KIT* and *KRAS* were almost exclusively mutated in Type II GCTs (Fig. 1a).

We also observed frequent mutations in chromatin remodeling genes, including five missense, one nonsense, and one frameshift deletion mutation in the Histone H3K4 methyltransferase *MLL2/KMT2D*. All seven *KMT2D* mutations were in YSTs. Besides *KMT2D*, another 20 chromatin remodeling genes were mutated in our cohort (Fig. 1a). These results highlight the potential role of epigenetic dysregulation in GCTs.

DNA repair genes were also frequently mutated in pediatric GCTs. Six tumors had mutations in *ATM*, a DNA damage response regulator. Five of six *ATM* mutations were found in YSTs. We also observed seven other DNA repair genes with mutations (Fig. 1a), including *PRKDC* (also known as DNA-PKcs). A recent report suggests that mutation of *PRKDC* might elevate DNA damage and mutation rate in cancer[30].

Lastly, we investigated known or suspected GCT driver genes in the validation study. We observed somatic protein-altering mutations in *DMRT1, CBL, NOTCH1* (three patients), *NOTCH2* (four cases), *EGFR*, *NRAS* (two cases), *PTEN* (two cases) and *MTOR*. We also observed recurrent mutations in genes of the Hedgehog signaling pathway, including *PTCH1* (three cases), *GLI1* (two cases), and *GLI2* (Fig. 1a). Supplementary Data 1 provides details of genes with somatic SNVs in multiple tumors, including scores of the predicted deleterious effect of variants. However, further studies will be required to fully assess the functional impact of these variants.

### DNA copy-number analysis

We assessed 148 GCTs for copy-number alterations with high density SNP arrays, using GISTIC 2.0[31] to determine significance. Type I tumors exhibited copy-number gains at 12p, 20q and 21 and losses at 1p and 6q; Type II tumors exhibited gains at 12p and 20q, and losses at 10 and 19q (Fig. 2a). GISTIC identified recurrent focal copy-number changes in several genes associated with germ cell development, GCT predisposition, and WNT signaling, including gain of *KRAS, ATF7IP, CCND2, DPPA3, GDF3, NANOG, LRP6, SOX18*, and *WNT5B*, and loss of *CXCL12, INSL3, NANOS3, SOX2, RET*, and *BTRC*.

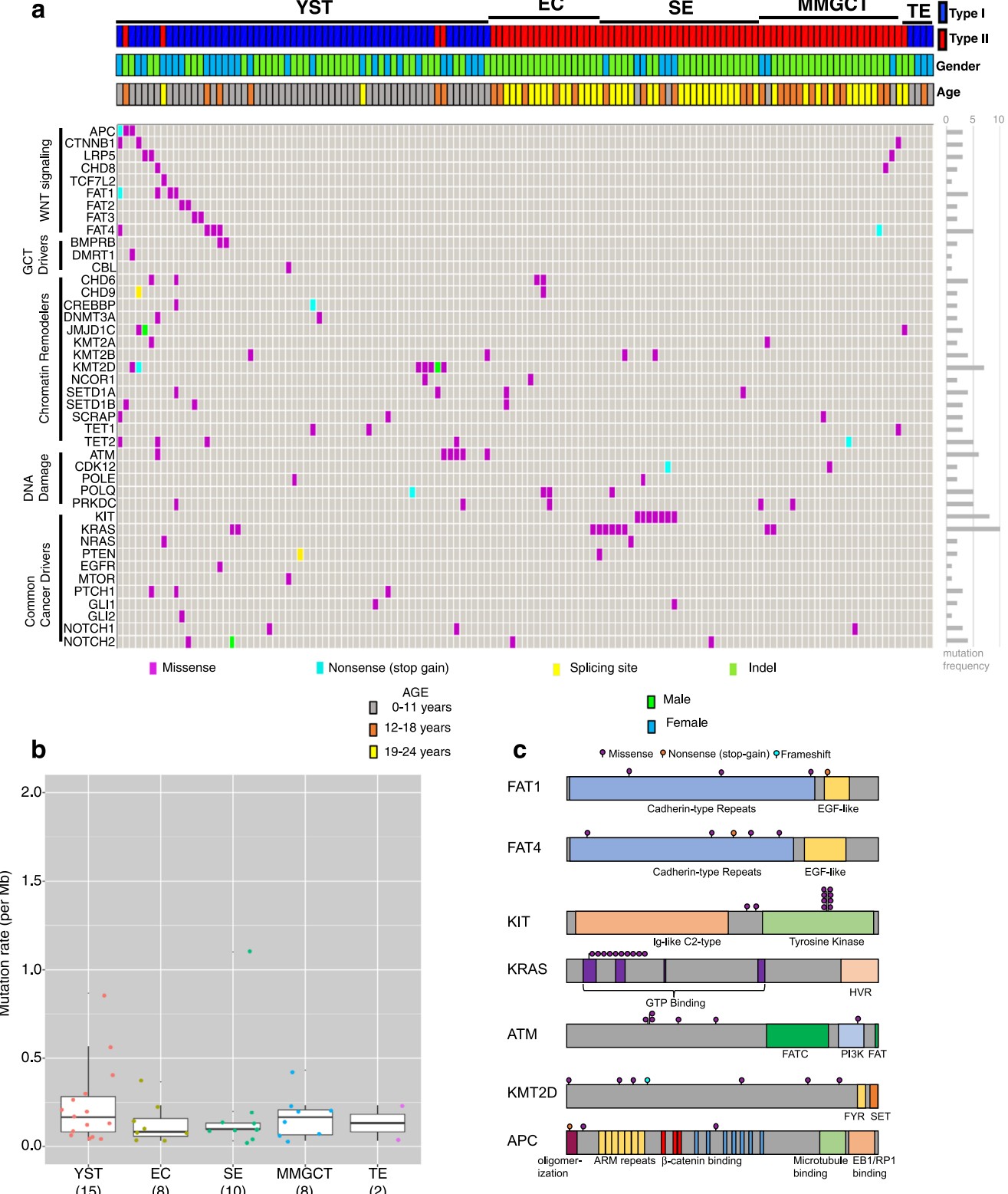

**Fig. 1 | Spectrum of somatic mutations and DNA copy-number alterations in GCTs. a** Somatic stop-gain, splice site, indel and missense mutations in 131 childhood and young adult germ cell tumors. EC embryonal carcinoma, SE seminoma/dysgerminoma, TER teratoma, MMGCT mixed malignant germ cell tumor, YST yolk sac tumor. **b** Boxplot of the number of somatic mutations per MB for each GCT subtype. The number of samples are marked below each histology type. The center line denotes the median value, the box contains the 25–75th percentiles and the whiskers mark the 5th and 95th percentiles. Values beyond these upper and lower bounds are outliers. **c** Schematic of coding sequence variants detected for selected cancer-relevant genes.

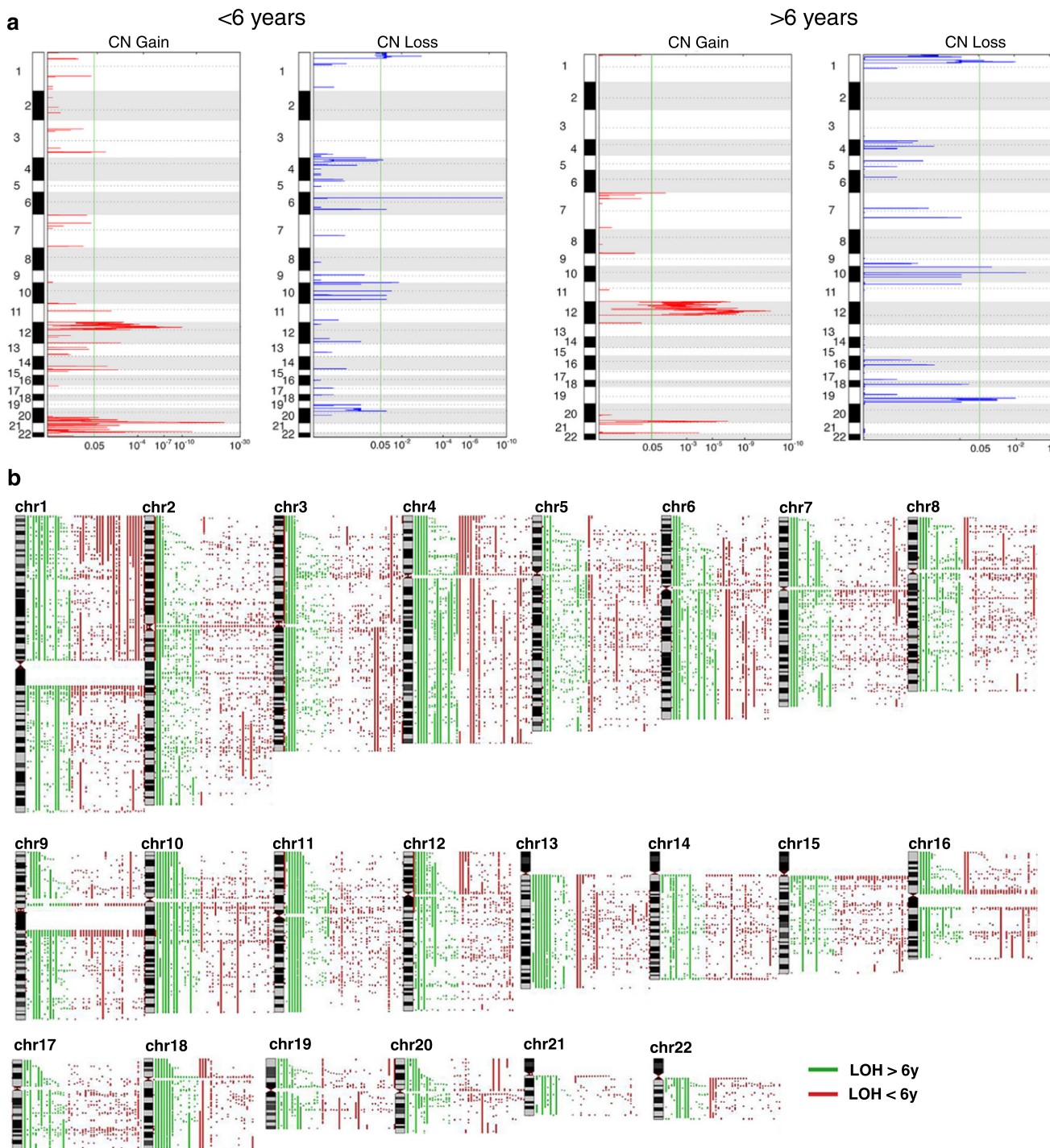

**Fig. 2 | Copy number changes and loss-of-heterozygosity in GCTs. a** GISTIC analysis of genome-wide copy-number variation in Type I and Type II GCTs. **b** Genome-wide loss-of-heterozygosity events in pediatric, adolescent and young adult GCTs. Loss-of-heterozygosity in GCTs from patients greater than (green) or less than (red) 6 years of age.

Recently, Taylor-Weiner and co-workers identified recurrent chromosome arm-level amplifications and reciprocal loss of heterozygosity as a major feature of adult testicular (type II) GCTs[19]. To separate type I from type II GCTs, we queried our SNP array dataset for LOH events and stratified the results by age (greater or less than 6 years of age[32]). We identified arm level LOH events in the majority of chromosomes of GCTs from patients older than 6 years. However, these large-scale LOH events were significantly less common in tumors from younger patients (Fig. 2b and Supplementary Fig. 2).

Figure 3 summarizes the pattern of somatic mutations and copy-number changes observed in the most frequently affected pathways in GCTs. As a complement to these analyses, we also performed whole-genome sequencing analysis in 10 tumor-normal pairs at 30× resolution. We used the DEFOR[33] and SCHALE[34] algorithms to assess copy-number changes, structural alterations and loss of heterozygosity (LOH) in the tumors. The results are shown in Supplementary Figs. 3–5. We observed recurrent somatic (tumor-specific) focal- and arm-level structural alteration events, recapitulating those described by lower-resolution array technologies in our study and by other groups, such as

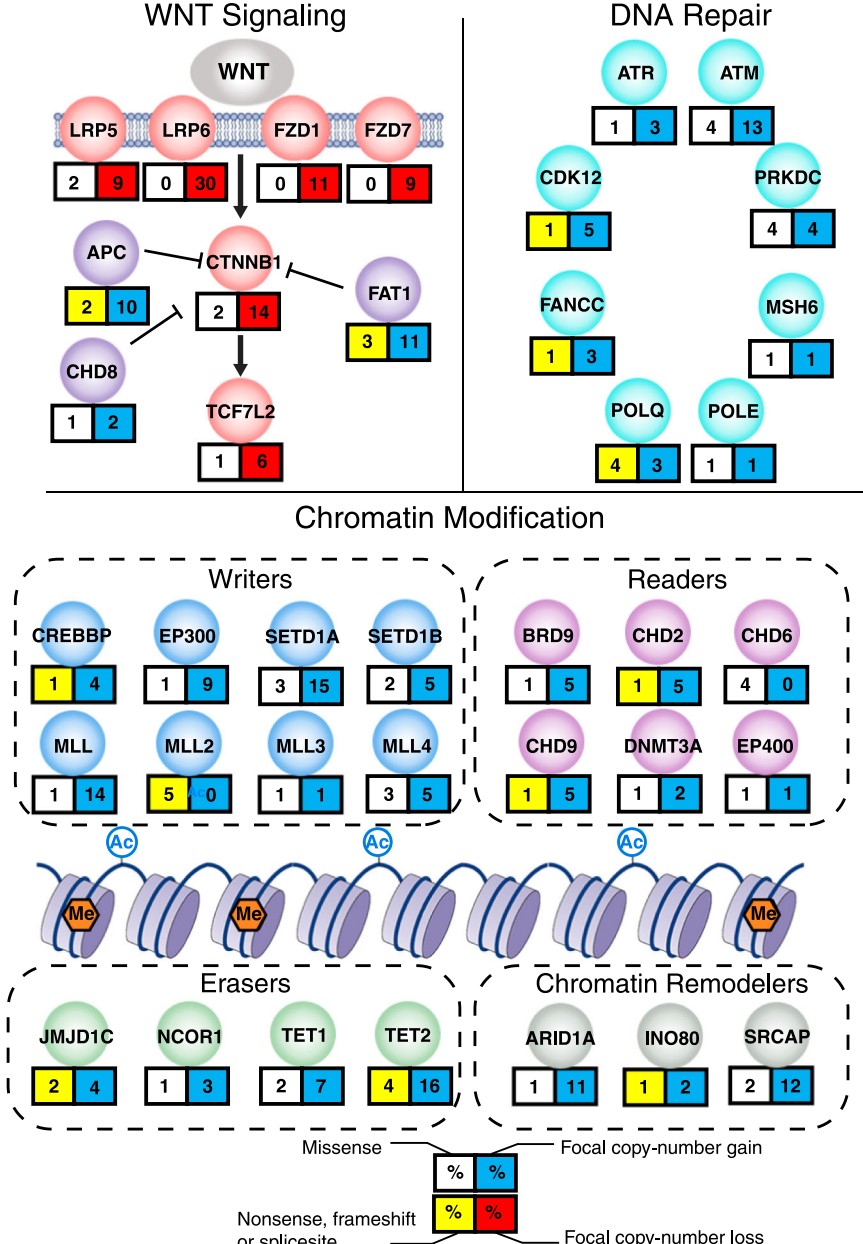

**Fig. 3 | Summary of somatic mutations and DNA copy-number alterations in WNT pathway, DNA repair and chromatin regulator genes in pediatric GCTs.** Selected pathways with recurrent somatic alterations in GCTs. For each gene, the percentage is based on the number of cases with SNVs (missense or nonsense/ frameshift/splice site), or focal copy-number gains or copy-number losses, as a proportion of the total number of samples analyzed for that type of alteration.

1p gain, 6q loss and 12p gain. Of note, an ovarian pure yolk sac tumor from a 23-year-old female did not exhibit any evidence of chromosome 12p gain (Supplementary Fig. 3), supporting the idea that such tumors are more closely related to Type I YSTs of young children. We also observed previously unreported copy-number changes and loss of heterozygosity (LOH) events. In addition, we analyzed RNA-Seq data using DEFUSE[35] to computationally identify possible gene fusions, which have not been reported previously (Supplementary Data 2). Further studies will be required to test the functional significance, if any, of these genetic alterations.

**Frequent DNA copy-number, promoter methylation and gene expression alterations of WNT pathway genes in type I and II tumors**

The occurrence of somatic mutations in WNT pathway genes in GCTs prompted us to examine the WNT pathway more closely by analyzing DNA copy-number, promoter methylation and gene expression data. A striking pattern emerged, with WNT pathway activators demonstrating low levels of promoter methylation and frequent focal copy-number gains, while repressors of WNT signaling display a reciprocal pattern, with high levels of promoter methylation and frequent focal copy-number losses (Fig. 4a, b; Supplementary Table 3). This pattern was present in type I and type II GCTs (both seminomas and non-seminomas), suggesting it may be a general feature of extracranial GCTs independent of age.

As an independent assessment of the effect of copy-number alterations on WNT pathway genes, we analyzed results from the TCGA study of testicular GCT[16]. Similar to pediatric GCTs, adult testicular GCTs showed a high frequency of tumors exhibiting focal copy-number gain of WNT activators, loss of WNT repressors, or both (Fig. 4c). Tumors exhibiting more than 5 such changes (designated WNT CN-rich, n = 85) had higher expression of beta-catenin compared

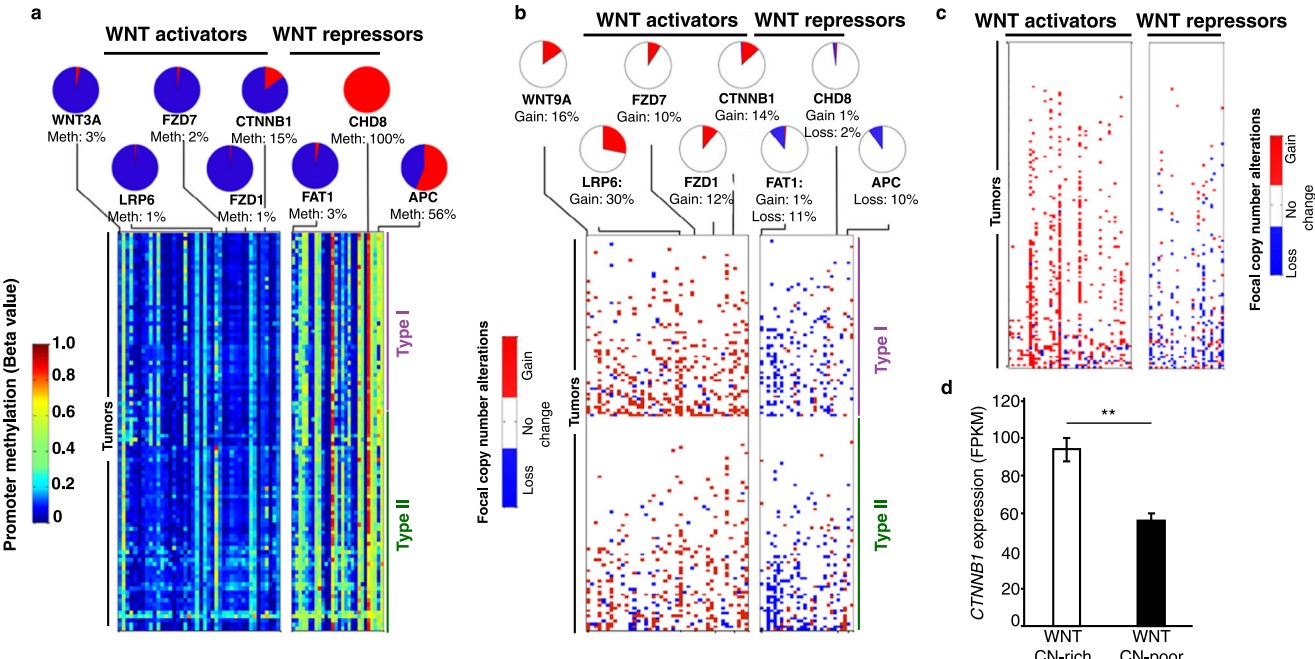

**Fig. 4 | Activation of the WNT pathway by copy-number alterations and methylation in GCTs.** Promoter methylation (**a**) and copy-number alterations (**b**) of genes predicted to activate or inhibit WNT signaling. **c** Copy-number alterations in WNT pathway genes in adult testicular GCT data from TCGA. **d** Beta-catenin expression in TCGA testis tumors stratified into tumors with >5 (WTN CN-rich; $n = 85$) and 0–5 (WNT CN-poor; $n = 71$) copy-number changes in WNT pathway genes. Data are presented as mean values ± SEM. Two-sided Student's $t$ test; **: $p = 0.001$.

to tumors with 5 or fewer changes in WNT pathway genes (WNT CN-poor, $n = 71$, Fig. 4d). To rule out the possibility that these patterns resulted from non-specific genomic instability in WNT CN-rich tumors, we compared the average numbers of genes exhibiting copy-number gains or losses in WNT CN-rich and WNT CN-poor tumors. The two groups did not exhibit significant differences (Supplementary Fig. 6).

### WNT pathway activity has prognostic significance in GCTs

Based on these observations, we predicted that GCTs would show evidence of active WNT signaling. We compared the expression level of six frequently used markers for WNT activation (*CTNNB1*/beta-catenin, *TCF1*, *TCF4*, *FZD7*, *MYC* and *CCND1*) in normal and tumor tissue. Compared to human PGCs[36] and normal testis, GCTs showed evidence of elevated WNT pathway activity, with highest levels in type I tumors (Fig. 5a,b).

To determine the possible prognostic significance of WNT pathway activation, we tested the association between WNT gene copy-number alterations (defined as gain of WNT activators or loss of WNT repressors) and outcome in our dataset. Patients whose GCTs harbored no focal WNT gene copy-number alterations experienced no relapses (Fig. 5c, left bar) and had 100% survival (Fig. 5d, black curve), while patients whose tumors had focal copy-number alterations of one to five WNT genes (WNT CN-poor group) had a slight but not significant increase in relapse (7% of patients) (Fig. 5c, middle bar) and a small decrease in survival rate (Fig. 5d, purple curve). However, in the third group of GCT patients with focal copy-number alterations of more than five WNT genes (WNT CN-rich group), we found a striking increase in occurrence of relapsed tumors (~30%, $p = 0.0007$, Fig. 5c, right bar) as well as a significantly decreased survival rate of patients (log-rank test, $p = 0.038$, Fig. 5d, orange curve). We obtained similar results using CN values of 3, 7 or 9 as the threshold value (Supplementary Figs. 7, 8).

To further test the association between WNT pathway activation and poor outcome, we evaluated an independent, previously described cohort of 108 non-seminomatous TGCT patients[37,38]. For three known WNT activator genes (*FZD1*, *FZD7*, and *CTNNB1*) with frequent

copy number gains as mentioned above, we observed significant associations between expression level and survival in this cohort (Fig. 5e). Taken together, these results suggest that aberrant activation of WNT pathway contributes to increased relapse and poor survival of GCT patients. The sample size of our dataset did not permit a separate evaluation of type I and type II tumors.

### Small molecule WNT inhibitors suppress the growth of GCT cells in vitro

The discovery of aberrant WNT pathway activation in GCTs has important translational implications, as several small molecule WNT inhibitors are in clinical development for treatment of cancer[39,40]. We treated GCT cell lines GCT44 and 1411H (YST), NTERA-2 (EC) and TCam2 (seminoma) with two different WNT inhibitors: the tankyrase inhibitor IWR-1 (Fig. 6a) and the PORCN inhibitor LGK-974 (Fig. 6b). Both inhibitors reduced the growth of GCT cell lines, with the largest effects in the YST cells.

Finally, we assessed the effect of WNT inhibitors in vivo. WNT signaling plays important roles in stem cells, including cancer stem cells[41–43]. The WNT target gene *PIWIL1*[44] confers stem cell fate and supports cancer cell growth[45,46]. *PIWIL1* is overexpressed in human GCTs[47], which exhibit features of impaired differentiation[48]. We previously reported that male zebrafish bearing mutations in the *bmpr1bb* gene develop testicular GCTs with gene expression similar to human GCTs[49]. The tumors exhibit impaired germ cell differentiation and elevated *piwil1* expression[21]. Using a *piwil1:eGFP* transgenic zebrafish reporter line[50] that permits live visualization of GCTs in *bmpr1bb* mutants (Supplementary Fig. 9), we tested the effects of WNT inhibition. We treated males with GCTs for 7 days of with DMSO vehicle control or with IWR-1. Treatment with IWR-1 led to a striking decrease of eGFP expression (Fig. 6c–e), indicating that WNT inhibition downregulates activity of the *piwil1* promoter.

Because *PIWIL1* promotes stem cell fate at least in part by inhibiting differentiation[45,46], we used two complementary assays to test whether the loss of eGFP signal was accompanied by evidence of increased differentiation in the tumors. First, histologic examination of

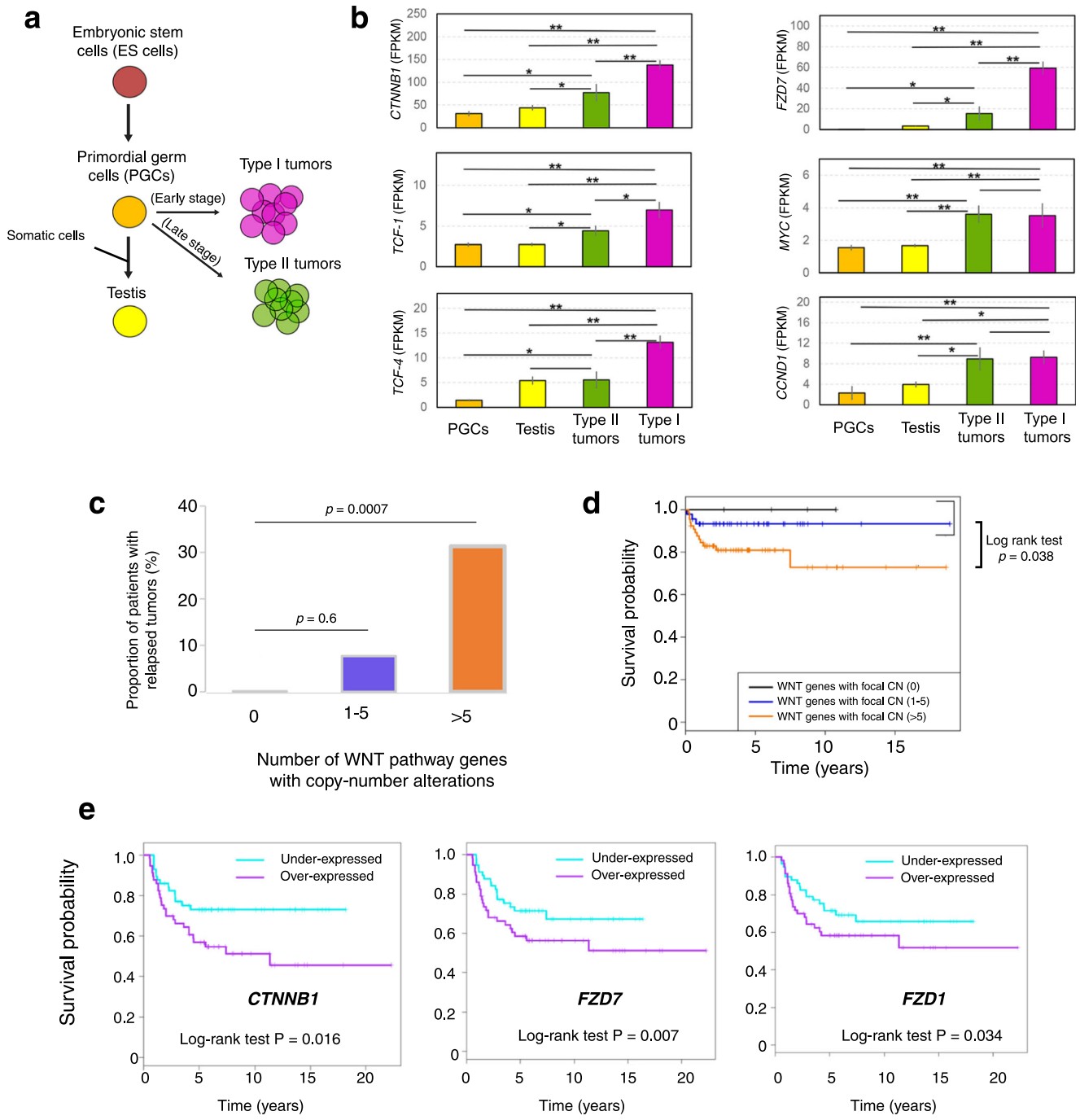

**Fig. 5 | WNT pathway alterations have prognostic significance in GCTs. a** Design of experiment. **b** Expression level of six marker genes of WNT pathway activity among PGCs, testis, type I and II GCTs (n = 32). Two-sided Student's t test p < 0.05 (*) and <0.001 (**); data are presented as mean values ± SEM. **c** Proportion of 114 GCT patients exhibiting relapse stratified by the number of WNT genes with copy-number alterations in the tumor; two-sided Student's t test. **d** Overall survival of 114 Type I and Type II GCT patients stratified by the number of WNT genes with copy-number alterations in the tumor. *: p = 0.038 by log-rank test. **e** Overall survival of an independent cohort[37,38] of 108 non-seminomatous Type II GCT patients stratified to high or low expression of the indicated gene.

H&E-stained tumor sections (Fig. 6f) showed that control DMSO-treated tumors consisted of sheets undifferentiated germ cells, with only scattered islands of mature spermatozoa, as we previously described[51]. In contrast, tumors from IWR-1 treated fish exhibited markedly more complete differentiation, with many lobules showing the full range of spermatocytic differentiation. Upon differentiating, germline stem cells enter meiosis. Therefore, we next used phospho-histone H2AX (pH2AX) as a marker of meiotic cells[52], we found that the WNT inhibitor-treated tumors exhibited increased pH2AX signal (Fig. 6g–i), in the characteristic clustered pattern of spermatocytes synchronously entering meiosis (Fig. 6h, inset). Thus, WNT inhibitor treatment interferes with the stem cell program in germ cell tumors and promotes differentiation of the tumor cells.

## Discussion

WNT signaling regulates cell migration, proliferation, differentiation, apoptosis and pluripotency[39]. Aberrant WNT signaling is associated with many types of human cancers[53]. Here we describe several convergent mechanisms predicted to drive WNT signaling activity in both type I and type II GCTs. Patients whose GCTs

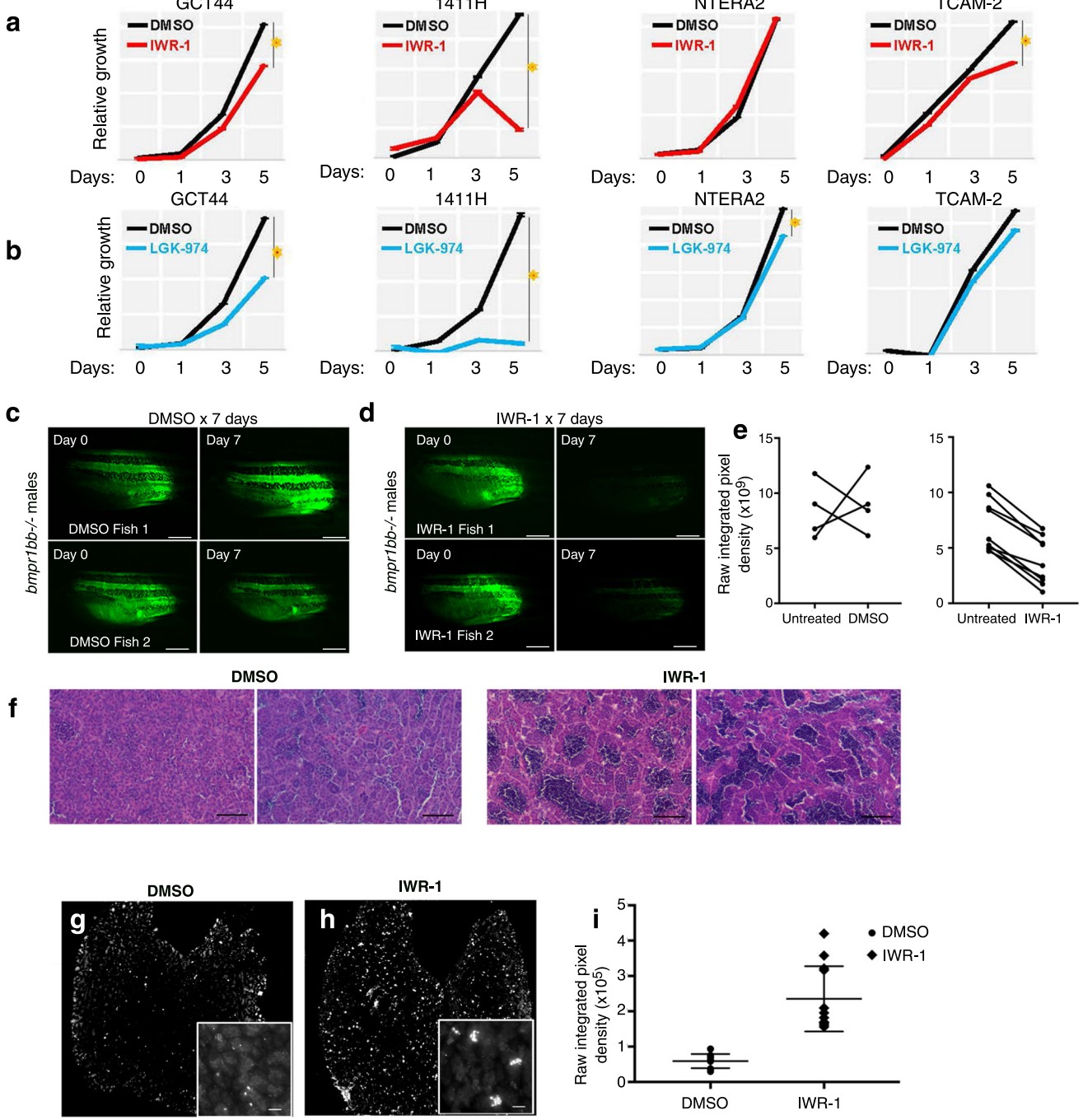

**Fig. 6 | The WNT pathway is active in GCTs and can be targeted by small-molecule WNT inhibitors. a, b** Relative growth of GCT cells exposed to tankyrase inhibitor IWR-1 (**a**) or PORCN inhibitor LGK-974 (**b**). Origin of cell lines is pediatric type I YST (1411H), a mixed type II nonseminoma combining EC and YST (GCT44), a pluripotent type II nonseminoma (NTERA2) and seminoma (TCam-2). **c, d** Effect of WNT inhibiton on expression of *piwil1-eGFP* reporter in *bmpr1bb−/−* males with testicular GCT. **c** Treatment with DMSO vehicle control (*n* = 4), (**d**), treatment with IWR-1 (*n* = 9). Scale bars: 2.5 mm. **e** Quantification of (**c**, **d**). *p* = 0.011 for the difference in mean percent change of fluorescence pre- and post-treatment by two-

sided *t*-test according to treatment arm. **f** Representative H&E-stained sections of GCTs from tumor-bearing zebrafish treated with DMSO control or with IWR-1. **g, h** Representative phosphohistone H2AX immunofluorescence of GCTs from zebrafish treated with DMSO (**g**) or IWR-1 (**h**). Inset: higher magnification view showing clusters of pH2AX-positive cells in IWR-1 treated tumors. Scale bar: 50 μm. **i** Quantification of (**g**, **h**). Data are presented as mean values ± SEM. For each condition (DMSO control or IWR-1 treatment), 3 sections from 4 different tumors were quantified (*n* = 12; *: *p* = 0.00000164 by two-sided *t*-test).

harbored frequent copy-number gains of WNT activators and loss of WNT repressors had significantly higher relapse and worse survival rates than those with fewer or no such changes. WNT activation was more common in type I tumors than in type II tumors, which may reflect their presumed origin from primed and naïve ESCs respectively[3]. However, we also observed significant

associations between high expression of WNT activators (e.g., *CTNNB1*, *FZD1*, and *FZD7*) and poor survival of type II non-seminoma GCT patients, suggesting WNT signaling may be a common driver of adverse outcome in GCT patients across age groups. Along with the recent description of gain of chromosome 3p25.3 in cisplatin-resistant non-seminoma tumors[20], the status of

WNT signaling may provide a useful tool for prognostic risk stratification in patients with GCT.

The WNT pathway has additional important links to the biology of GCT. For example, the miR-371~373 microRNA cluster is a well-known biomarker for clinical diagnosis of GCT[54,55]. A previous report demonstrated that WNT pathway activity can upregulate miR-371~373[56], a pluripotency cluster highly expressed in YSTs, providing a possible explanation for the upregulation of miR-371~373 in at least some GCT patients.

Most importantly, this resource can help support the development of targeted therapeutic strategies for GCT patients. Several small-molecule WNT inhibitors are being developed as anticancer therapeutics[39]. We showed that two small molecule WNT inhibitors targeting distinct nodes of WNT signaling (LGK-974 and IWR-1) can significantly suppress growth of GCT cells in vitro, and promote differentiation of tumor cells in vivo. Further studies are required to understand if the effects of WNT inhibition result from similar or different mechanisms in seminomatous and non-seminomatous GCTs. Nevertheless, these results suggest that inhibition of aberrantly active WNT pathway may be a promising therapeutic strategy to improve survival of GCT patients.

## Methods

### Sample collection
A total of 229 patients with germ cell tumors (GCTs) were enrolled in this study. Tumor samples and clinical information used in this study were obtained under informed consent and approval by the Institutional Review Board of the participating facility. Samples were assembled from collections at the University of Texas Southwestern Medical Center, Dallas, TX USA; Children's Oncology Group; Boston Children's Hospital, Boston, MA USA; the Erasmus University Medical Center, Rotterdam, Netherlands; and the Hospital Sant Joan de Déu, Barcelona, Spain. All samples were de-identified at the source. Genomic DNA and RNA were extracted using the QIAamp DNA Mini kit (Qiagen) or Gentra PureGene kit (Qiagen) and the RNeasy Mini kit (Qiagen), respectively.

### Whole-exome sequencing and variant calling
Exome capture was carried out using SureSelect Human All Exon v4+UTRs (Agilent Technologies), and sequencing was performed with a HiSeq 2000 instrument (Illumina) with 100 bp paired-end reads to a mean coverage of 130× for exomes. Raw reads were mapped to human reference genome (hg19) using BWA[57]. Duplicates were removed using Picard (http://picard.sourceforge.net/). We performed base quality recalibration prior to variant calling, and used read quality score >30 in variant calling[58]. Matched tumor-normal BAM files were used as input for VarScan software[59] to identify somatic single-nucleotide variants (SNVs) and small-scale insertion/deletions (INDELs). Variants were identified based on (1) variant allele frequency (VAF) in the tumor ≥ 10% and VAF in normal is 0%; (2) variants must be supported by both strand and >6 reads; (3) mapping quality ≥30 and base quality ≥15; (4) both tumor and normal should be covered by ≥10 reads in variant loci. In addition, variants with overall alternate allele frequency >0.01 in the 1000 Genome Project database or in the National Heart, Lung, and Blood Institute (NHLBI) Exome Variant Server (ESP6500) database were also excluded. Consensus oncogenic genes and variants were identified on the basis of the Catalogue of Somatic Mutations in Cancer (COSMIC) database. Detected variants were annotated on the basis of CCDS, RefSeq, UCSC and Ensembl annotations. All candidate variants were visually inspected in the Integrative Genomics Viewer (IGV) genome browser[60] to exclude likely germline mutations and sequencing artifacts.

### Targeted deep sequencing
High-throughput targeted sequencing by multiplex PCR was performed on whole genome–amplified DNA. The custom panel and primer pairs for targeting coding regions of 66 genes were generated by Qiagen. An equimolar pool of all PCR products was sequenced on the MiSeq instrument (Illumina), with paired-end 2 × 150 reads. The GeneRead target enrichment panel variant calling pipeline (http://ngsdataanalysis.sabiosciences.com/NGS2/) is optimized to perform data analysis for the custom deep-seq panels from Qiagen, which was used for base calling and alignment of reads to the reference human genome and variant calling. Given low mutation rate in GCTs, only 16 somatic protein-altering mutations (from 52 of 133 GCTs) called from whole-exome sequencing analysis were covered in the targeted deep sequencing panel. 100% (16 of 16) of validation rate by deep-seq panel was observed.

### Whole-genome sequencing
Genomic DNA from 10 tumor-normal pairs was subjected to standard Illumina paired-end DNA library construction. Adapter-ligated libraries were amplified by PCR and subjected to DNA sequencing using the HiSeq platform (Illumina) according to manufacturer's instructions. Target coverage was >30×. We mapped raw sequencing reads to the human reference genome (version hg38) using BWA algorithm[57], with base qualities of the aligned reads recalibrated and realigned using GATK[58]. Copy-number changes, structure alterations and loss of heterozygosity (LOH) calling was performed by DEFOR[33] and SCHALE[34] algorithms following the default settings.

### RNA sequencing
RNA of 32 GCT samples was sequenced on Illumina HiSeq2000 according to the manufacturer's protocol (Illumina). 100-bp paired-end reads were assessed for quality and reads were mapped using CASAVA (Illumina). The generated FASTQ files were aligned by Bowtie2[61] and TopHat2[62]. Cufflinks[63,64] was used to assemble and estimate the relative abundances of transcripts at the gene and transcript level. DEFUSE[35] was used for fusion gene discovery.

### Copy number and LOH analysis
Genomic DNA from 148 GCT samples was analyzed by SNP array technologies using the Illumina Omni 2.5 M SNP array and Affymetrix OncoScan array, according to the manufacturers' recommendations. Illumina array raw data were processed and normalized in GenomeStudio (Illumina). For Affymetrix arrays, CEL files were generated to contain raw data. To minimize bias in processing data from two platforms, we chose to use Nexus Copy Number Discovery 7.0 software (BioDiscovery, Inc.), which can process raw data from both platforms with the same algorithm and procedure. In this software, the data were corrected for GC content and segmented by using SNP-FASST2 algorithm with default parameters. Human genome assembly GRCh36/ hg18 was used as a reference and then was lifted over to hg19. Nexus Copy Number Discovery 7.0 software was used with recommended settings to determine Loss-of-Heterozygosity (LOH) regions in each sample, using raw data from the Illumina Omni 2.5 M SNP array. Data were normalized in GenomeStudio (Illumina). Genomic Recurrent Event ViEwer (https://www.well.ox.ac.uk/GREVE/) was used to plot the figure.

### Methylation array
Prior to methylation analysis, 500 ng genomic DNA was treated with sodium bisulfite using the EZ DNA Methylation Kit (Zymo Research, Orange, CA) according to the manufacturer's protocol. Genome-wide methylation analysis was performed using the Infinium Human-Methylation450 BeadChip array (Illumina, San Diego, CA) in the University of Minnesota Genomics Center following Illumina's standard protocol. For FFPE samples, we used the Infinium FFPE DNA Restore Kit (Illumina, Inc.) to repair DNA prior to array methylation analysis. All DNA samples were assessed for quality prior to analysis and duplicates were included for 19 samples to control for chip variation.

Raw intensity (idat) files were converted by using the methylumi package[65]. Combined with IMA package[66], DNA methylation sites with missing values, cross hybridizing probes, located within repeat regions or on sex chromosomes were excluded, resulting in a total of 392,714 probes retained. Methylation data were subsequently converted into β values, ranging from 0 (unmethylated) to 1 (fully methylated), and these values were normalized using a beta-mixture quantile normalization method (BMIQ)[67].

## Gene expression analysis

Association between gene expression and survival was calculated based on 108 GCT cases measured by Affymetrix U133A microarray platform (Korkola JE et al., 2009; Korkola JE et al., 2015). Signal intensity CEL files were downloaded from Gene Expression Omnibus (GEO) repository at http://www.ncbi.nlm.nih.gov/geo/, data set GSE3218 and GSE10783. CEL files were then processed by Affymetrix Power Tools (APT) with Robust Multiarray Average (RMA) method. Cox proportional hazards model was used to calculate the statistical significance, as well as hazard ratios and 95% confidence intervals of the associations between the gene expression and survival. Kaplan-Meier curves were generated based on gene expression values dichotomized into over- and under-expressed groups using the within cohort median expression value as a cutoff.

## Cell culture, drug treatments, and cell growth assay

Human Germ Cell Tumor Cell line NTERA2 was acquired from the American Type Culture Collection (ATCC). Tcam-2 cells were the gift of Dr. Sohei Kitazawa (Kobe University, Japan). GCT44 and 1411H were provided by Drs. Nick Coleman and Matthew Murray (University of Cambridge, UK). All cell lines were validated by STR genotyping and determined to be mycoplasma-free using the Mycoalert detection kit (Lonza) monthly according to the manufacturer's instructions. Cells were cultured DMEM supplemented with 10% FBS in a 37 °C humidified incubator containing 5% $CO_2$. For cell growth assays 1411H cells were trypsinized with 0.25% Trypsin EDTA (GIBCO) and the remaining lines were trypsinized with 0.05% Trypsin EDTA (Sigma-Aldrich). After trypsinization cells were replated in triplicate at 35,000 cells per well in 12-well cell culture plates (Costar) in sets of 8 plates per line. 24 h after plating half of the plates were treated with either various concentrations of PORCN inhibitor WNT974 (0 μM, 1.0 μM, 2.5 μM, 10.0 μM) or TNKS inhibitor IWR1 (0 μM, 1.0 μM, 5.0 μM, 10.0 μM). Media and drug were replaced daily for 5 days. Plates were collected on days 0, 1, 3, and 5 by aspiration of media followed by a wash with Dulbecco's PBS (Sigma-Aldrich) and addition of 10% formalin. At the end of the collection period plates were washed once in water and stained with 0.1% crystal violet solution for 20 min. Following staining plates were washed 3 times with water and allowed to air dry. Crystal violet was solubilized in 10% acetic acid with shaking and absorbance read at 590 nm.

## Zebrafish GCT treatment and immunofluorescence

*Danio rerio* were maintained in an Aquaneering aquatics facility according to industry standards. All work was performed under protocols approved by the Institutional Animal Care and Use Committee at UT Southwestern Medical Center, and AALAC-accredited institution. The *bmpr1bb* and *piwl1-eGFP* strains were previously described[21,50]. For drug exposure experiments, *piwl1-eGFP; bmpr1bb*[−/−] males with testicular GCTs and *piwl1-eGFP; bmpr1bb*[+/+] controls were treated with DMSO vehicle control or 1 μM IWR-1 for 7 days. Fresh drug was added daily. Images of the tumors were taken at the beginning and end of the treatment in a stereodissecting microscope equipped for epifluorescence, and the raw integrated pixel intensity in the fluorescence channel was measured with ImageJ[68]. The maximal permitted tumor size of 1000 mm3 was not exceeded. At the end of the treatment, fish were euthanized, and histologic sections prepared. Immunofluorescence for phosphohistone H2AX was performed as described[21] and the results quantifies as above with ImageJ.

## Statistics and reproducibility

Statistical tests were two-sided Student's *t* test or log-rank test, as applicable. All experiments involving cell lines or animal models were replicated at least twice with similar results. In Fig. 6f, micrographs shown are representative images from 4 DMSO-treated and 4-IWR-1-treated tumors. In Fig. 6g–h, immunostains shown are representative images from 4 DMSO-treated and 4-IWR-1-treated tumors.

## Reporting summary

Further information on research design is available in the Nature Portfolio Reporting Summary linked to this article.

## Data availability

Genomic sequencing data of germ cell tumor samples in this study are deposited to dbGaP with Accession Number phs002009.v1.p1. Tumor methylation data are deposited in the Gene Expression Omnibus (GEO) repository under Accession GSE183798. The following databases were used in the analysis: 1000 Genome Project database (https://www.internationalgenome.org); National Heart, Lung, and Blood Institute (NHLBI) Exome Variant Server (ESP6500; https://evs.gs.washington.edu/EVS/); and the Catalogue of Somatic Mutations in Cancer (COSMIC). Source data are provided with this paper.

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

## Acknowledgements

We thank the patients and families who contributed biospecimens to this study. Supported by grants from the National Cancer Institute (CA135731 to J.F.A., CA151284 to J.N.P., T32 CA099936 to J.B. and U10180886, U24CA114766 and U10CA180899 supporting the Children's Oncology Group), A St. Baldrick's Foundation Consortium award (to A.L.F., M.D.K., and J.F.A.) and the Cancer Prevention and Research Institute of Texas (RP110394 to J.F.A.). J.M. acknowledges the contribution of "Asociación Anita" for the work on GCT at Hospital Sant Joan de Déu, Barcelona.

## Author contributions

L.X., J.L.P., A.S., J.W.O., L.H.J.L., J.M., J.N.P., B.W.D., A.L.F., J.F.A. conceptualized the work; L.X., K.S.C., A.B., M.D.K., F.S., X.X., L.G., D.B., F.P., Y.X., L.T., D.R. investigated, curated and provided formal analysis of data; L.X., J.L.P., A.S., A.A.S., N.J.F., S.S., J.B., K.B., A.J.M.G., J.S. were involved in experimental investigation and the development of animal models. L.X., J.L.P., A.S., J.F.A. prepared figures. L.X. and J.F.A. wrote the paper and all authors commented on the final paper. J.B., J.N.P., J.M., and J.F.A. obtained funding.

## Competing interests

A.L.F.: Clinical Advisory Board, stock and compensation from Decibel Therapeutics. The remaining authors declare no competing interests.

## Additional information

[1]Quantitative Biomedical Research Center, University of Texas Southwestern Medical Center, Dallas, TX, USA. [2]Department of Population & Data Sciences, Peter O'Donnell Jr. School of Public Health, University of Texas Southwestern Medical Center, Dallas, TX, USA. [3]Department of Pediatrics, University of Texas Southwestern Medical Center, Dallas, TX, USA. [4]Department of Urology, University of Texas Southwestern Medical Center, Dallas, TX, USA. [5]Department of Urology, University of California San Diego, San Diego, CA, USA. [6]Department of Preventative Medicine, University of Southern California Keck School of Medicine, Los Angeles, CA, USA. [7]Children's Oncology Group, Monrovia, CA, USA. [8]The Hospital for Sick Children, University of Toronto, Toronto, ON, Canada. [9]Riley Hospital for Children, Indianapolis, IN, USA. [10]Department of Pediatrics, Yale University School of Medicine, New Haven, CT, USA. [11]Department of Pediatrics, University of Minnesota, Minneapolis, MN, USA. [12]Princess Máxima Center for Pediatric Oncology, Utrecht, The Netherlands. [13]Department of Bioinformatics, University of Texas Southwestern Medical Center, Dallas, TX, USA. [14]Department of Pathology, Boston Children's Hospital, Boston, MA, USA. [15]Sant Joan de Déu Barcelona Children's Hospital, Barcelona, Spain. [16]Department of Pathology, University of Texas Southwestern Medical Center, Dallas, TX, USA. [17]Department of Molecular and Cellular Biology, University of California Davis, Davis, CA, USA. [18]Dana-Farber/Boston Children's Cancer and Blood Disorders Center, Boston, MA, USA. [19]Cancer and Blood Disease Institute, Children's Hospital Los Angeles, Los Angeles, CA, USA. [20]Department of Pediatrics, University of Southern California Keck School of Medicine, Los Angeles, CA, USA. [21]Department of Medicine, University of Southern California Keck School of Medicine, Los Angeles, CA, USA. [22]Present address: Blank Children's Hospital, Des Moines, IA, USA. [23]These authors contributed equally: Lin Xu, Joshua L. Pierce, Angelica Sanchez. ✉e-mail: lin.xu@utsouthwestern.edu; jamatruda@chla.usc.edu

