## [Peer Review File · Nature Communications]

Integrated genomic analysis reveals aberrations in WNT signaling in germ cell tumors of childhood and adolescenceREVIEWER COMMENTS

Reviewer #1 (Remarks to the Author): Expert in GCTs

Reviewer Comments to the Author:

In the article "Integrated genomic analysis reveals aberrations in WNT signaling in germ cell tumors of childhood and adolescence" Xu et al, performed integrative genomic analysis of germ cell tumors of childhood and adolescence. The data fills a gap of knowledge and it is of interest for the clinical management of this cancer type. It has potential to aid classification for tumors and to propose new therapies

I have however several comments and questions that I hope will help improving the manuscript and highlighting the key results.

1) One of the main hypothesis stated by authors was a different biological nature of prepubertal and post pubertal CGTS. In the abstract the authors state "These results highlight the distinctive mechanisms underlying the development of childhood and adolescent GCTs and provide a foundation for future efforts to develop targeted therapies for these cancer", however the main results derived from the study is a general involvement of WNT pathway activation seen in both pre-pubertal and post pubertal GCT. So, the above conclusion is not totally supported by these results and to my opinion should be rephrase accordingly. A slight difference in the LOH status of cases younger than 6 years old and older than 6 years old is seen in the data but those results do not seem strong enough to drive such a conclusion. If the authors refer to wnt pathway mutations in type I vs KIT/KRAS mutations in type II, then maybe the expression level of MAPK pathway should be addressed in these tumors.

3) the manuscript has potential to help in the classification of extracranial GCTs. Currently classification of testicular GCTs (WHO2016 guidelines) and ovarian GCTs (WHO 2014 guidelines) does usually not follow the same criteria. Have the authors performed correlation of the molecular data with sex? Figure 1 shows a snapshot of molecular results for the study, adding age and sex would complement the correlations and improve the results. Did the authors find differences with those cases older than 18 years old? The focus of the study were childhood and adolescents but then the results get sort of confusing in the comparisons performed.

4) How were missense mutations interpreted as pathogenic? No criteria are stated in the methods. This is important to justify for example pathogenicity in APC variants. In cases in which a wnt repressor is mutated, were those mutations accompanied by a second hit? How were the wnt repressors and activators list chosen? Can the authors provide stats for the methylation clustering?

5) Regarding WNT pathway alterations, the results are quite streaking and have a lot of potential as the authors used both in vitro and in vivo models to prove a plausible effect of wnt inhibitors as a treatment. The mutations in wnt genes correlate mostly to YST subtypes, even in mixed tumors with a YST component. In intracranial GCTs genomic analysis, driver mutations in mixed tumors were the same in both histological components of the tumor, is this the case in extracranial GCT? One would expect given the results of this study that, in this case, is limited to the YST component. This could be assessed by microdissection the tissue or inferred from the tumor relative content and VAF of the mutation. Do other alterations in wnt pathway ie hypermethylation of repressors or CN gains of activators correlate to a specific subtype? Sex? Or group age that could aid in personalizing the use of these therapies?. In vitro data with wnt inhibitors shows a stronger effect in lines with a YST component.

The authors used TCGA testicular dataset to validate their results but only chose 3 wnt genes, is there a reasons? Were the other wnt genes analyzed?

6) In general, a more detailed methods section (could be included in supplements) will help in the reading and the interpretation. There are multiple examples throughout the text, i.e. the number of cases with CN rich or CN poor is not stated anywhere, and all data is given in percentages. This applies to both cases studied in this manuscript and cases from TCGA. How were gene lists chosen or why a CN of 5 was the chosen threshold is not justified, neither a range of CN observed is provided. Were multiple comparisons considered? Fig 3 is another example> what do the % in the boxes correspond to? Number of missenses mutations in each gene or number of cases mutated in a given gene?

7) the authors have germline data. Are there plausible candidate genes that could increased the

risk to develop GCTs in childhood?

8) Tumor mutation burden was low in this sample set. However, the authors report several mutations in DNA repair genes. Therefore, the TMB in some of those cases (POLE, PRKDC) should be much higher (POLE is hypermutated tumors) so to drive conclusion about pathogenicity for those variants a mutational signature analysis or individual TMB analysis should be performed. Were there correlations between DNA repair mutations and cisplatin resistance?

8) Did the authors explore a link between mutations in chromatin remodelers and alteration in wnt genes methylation status?.

9) It is striking the number of mutations found in FAT genes. FAT genes functions are still unclear although FAT1 is known to activate wnt pathway and other roles in development have been postulated. Is CTNNB1 highly expressed and nuclear in FAT2, 3 mutated cases? FAT4 is the closest to the drosophila fat gene and therefore might be implicated in hyppo pathway. Were other hyppo genes mutated in type I or II GCTs. The authors say that FAT 4 is mutated in type II tumors but the Figure 1A does not show that correlation, could this be a mistake?

Reviewer #2 (Remarks to the Author): Expert in computational biology

Several molecular studies suggested distinct molecular origins and developmental mechanisms of type I and type II Germ cell tumors (GCTs). In this study authors established the activation of the WNT pathway as a molecular signature of both type I and type II GCTs. Authors have investigated several genomic aspects of GCTs—somatic mutations, copy-number alteration, and differential promoter methylation—in relation WNT pathway. In addition, they have also shown that WNT inhibitors can suppress GCT cells both in vitro and in vivo suggesting novel targeted therapies. In general, the design of experiments and data analysis were well-planned, covering most of the key aspects in integrative genomic studies. Addressing the following comments below would further strengthen and better comprehend the scientific advances.

1. What fraction of the whole-exome sequencing and custom-capture deep sequencing overlap with RNA-Seq data? Authors reported stop-gain mutations in APC and FAT1 that likely result in truncated proteins. Do authors find any truncated transcripts in the RNA-Seq data?
2. Did authors perform any base quality recalibration prior to variant calling? Authors used a read quality filter of ≥ 15 . However, score of 20 (99%) or 30 (99.9%) are typically recommended.
3. A significant fraction of the analysis is based on the WNT CN-rich and WNT CN-rich classification. What is the rationale of using "5" as a hard threshold? Did authors try other cutoffs? If so, how did they choose "5"? A note of this in the methods section may benefit other researches performing similar analyses.
4. Authors proposed/reported activation of the WNT pathway is associated with poor clinical outcomes in both children and adolescents. However, the analysis is not performed independently for type I and type II.

Minor:

1. Introduction: novels therapies -> novel therapies
2. Results: Consider rephrasing "The 51 genes were ..."
3. Methods: Raw reads was mapped -> Raw reads were mapped

Reviewer #3 (Remarks to the Author): Expert in pediatric oncology

The manuscript "Integrated genomic analysis reveals aberrations in WNT signalling in germ cell tumors of childhood and adolescence" presents a range of sequencing analyses from a cohort of germ cell tumours and presents evidence to suggest that in at least some of these, WNT signalling activation is the driver mechanism.

The major strengths of the manuscript are the presentation of the genomic features of a tumour type that has, in paediatric cases at least, been relatively understudied. The manuscript also presents a functional analysis of the contention that WNT activation is a driver in a proportion of these tumours.

The weaknesses are that the more comprehensive sequencing approaches (WES and RNA seq) are applied only to a small number of the cohort, and the majority of samples do not have exome sequencing or RNA sequencing. This will necessarily restrict the potential to identify the full range of variants that might be observable in the tumours. In addition, beyond the assessment of copy number variants, there appears not to be any other analysis for other structural variants, such as fusions, which might be available from RNA data. Thus whilst I am enthusiastic about a manuscript addressing the genomic variants in less-studied tumour types, my concern is that the data presented in this paper is insufficiently comprehensive and generalizable.

Listed below are more specific questions and comments for the authors.

1. Whilst I appreciate the significant work involved in this suggestion, if there were the opportunity to expand the WES and RNAseq in the cohort, that would be highly desirable.
2. With the variants described in the WNT genes in Figure 1 and the supplemental information, what evidence supports these mutations as pathogenic or driver mutations? Indeed, this same question could be addressed for the non-synonymous and splice variants (other than established hot-spot mutations) identified in the cohort. The point here being to distinguish, in some way, true driver mutations across the cohort.
3. In Figure 2, why was the age cut-off selected to distinguish type I from type II GCTs when the histology is available? In table 1 there seem to be a number of YST in patients >6 yrs, which suggests that age segregation might not be completely accurate. Further, what distinct molecular origins are the authors proposing that give rise to type I and type II GCT? Are there other data they have which might support these origins? Is the suggestion that the significant association of whole chromosome LOH with the <6yo group indicates an origin of a mis-segregation event followed by chromosome duplication? Is this supported by the karyotype data? Are there mutations in these tumours that are associated with this mechanism? Are the LOH copy neutral or associated with copy number gain or loss?
4. I could not see, in the main body of the manuscript, that the authors distinguished focal copy number losses as heterozygous or homozygous. Similarly, I could not determine if copy number gains were quantitated in any way.
5. Did the authors look for other types of structural variants?
6. The WNT observation is, I agree, intriguing. The gene expression data was confusing as it was not clear whether expression data derived from this cohort (32 RNAseq samples) was used to define beta-catenin expression. Was this RNA data used to look more broadly at gene expression? I did not understand why a cut-off of 5 or more copy number alterations was chosen. At first glance, this seemed quite arbitrary. The relationship between 5 or more CNV and beta catenin expression also seems a little unclear. Why would 5 or more copy number changes be required to elevate beta-catenin expression? What is the effect of each of the copy number changes on the expression of that individual gene? Are these heterozygous or homozygous copy number losses? Is there evidence to support haploinsufficiency of these genes as a mechanism of WNT pathway activation?
7. In reference to data in Figure 5, the authors state "To determine the possible prognostic significance of WNT pathway activation, we tested the association between WNT gene copy-number alterations (gain of WNT activators and loss of WNT repressors) and outcome in our dataset". Are these the same CNV that are described in Figure 4?
8. Is the PIWIL1 gene subject to CNV in GCT? Why isn't this gene included in the data in Figure 4 or 5?

Reviewer #4 (Remarks to the Author): Expert in zebrafish cancer models

Key Results:

In this manuscript the authors use zebrafish to determine the role of Wnt signaling in GCT progression. They utilized a previously established zebrafish model of GCT which has a loss-of-function mutation in *bmpr1bb* leading to impaired spermatogonial differentiation and testicular tumours.

The authors used adult male *bmpr1bb*; *piwil1:EGFP* fish in which GCT tumours had developed which were then treated with 1 μ M of IWR-1 for 7 days. In the treated fish they observed a loss in EGFP fluorescence and an increase in pH2AX. These results suggest that inhibition of Wnt signaling is causing differentiation of the GCT cells as pH2AX is a marker of meiosis.

Originality, significance, validity and conclusions:

The zebrafish results are an elegant addition to the manuscript, providing *in vivo* data to support the bioinformatic, epidemiological and *in vitro* data.

Statistics:

(Figure 6 only)

Overall the statistics used are appropriate. However, the number of fish and any experimental replicates should be included in the legend.

Referencing:

Appropriate.

Clarity and context

Well written.

Suggested Improvements

1) The authors propose that the GCT tumours treated with IWR-1 are more 'differentiated' but this is based only on the pH2AX staining. Given that this is a key outcome from the study it would be beneficial to have further evidence to support this claim.

Perhaps H&E staining of the control vs IWR-1 treated gonads could reveal histological changes supporting this claim?

2) It is not clear from the manuscript what the significance is for the loss of *piwil1:EGFP* expression in IWR-1 treated *bmpr1bb*^{-/-} mutants vs no change in WT animals (Fig 6F). Presumably as *piwil1* is Wnt-regulated and upregulated in GCT this is a control for the efficacy of IWR treatment? Initially I thought *piwil1:EGFP* expression was used as a readout of tumour size and/or differentiation status, but I am not sure if this was the authors intent. This should be clarified.

3) Panels 6C and 6D should be labeled to indicate that the fish were *bmpr1bb*^{-/-}, it would also be valuable to include the *piwil1:EGFP* expression in WT fish for comparison.

NCOMMS-20-15650A

“Integrated genomic analysis reveals aberrations in WNT signaling in germ cell tumors of childhood and adolescence”

Response to reviewers

REVIEWER COMMENTS

Reviewer #1 (Remarks to the Author): Expert in GCTs

Reviewer Comments to the Author:

In the article " Integrated genomic analysis reveals aberrations in WNT signaling in germ cell tumors of childhood and adolescence" Xu et al, performed integrative genomic analysis of germ cell tumors of childhood and adolescence. The data fills a gap of knowledge and it is of interest for the clinical management of this cancer type. It has potential to aid classification for tumors and to propose new therapies

I have however several comments and questions that I hope will help improving the manuscript and highlighting the key results.

1) One of the main hypothesis stated by authors was a different biological nature of prepubertal and post pubertal CGTS. In the abstract the authors state “These results highlight the distinctive mechanisms underlying the development of childhood and adolescent GCTs and provide a foundation for future efforts to develop targeted therapies for these cancer”, however the main results derived from the study is a general involvement of WNT pathway activation seen in both pre-pubertal and post pubertal GCT. So, the above conclusion is not totally supported by these results and to my opinion should be rephrase accordingly. A slight difference in the LOH status of cases younger than 6 years old and older than 6 years old is seen in the data but those results do not seem strong enough to drive such a conclusion. If the authors refer to wnt pathway mutations in type I vs KIT/KRAS mutations in type II, then maybe the expression level of MAPK pathway should be addressed in these tumors.

- Response: We agree with the reviewer that the data distinguishing Type I and Type II tumors, while intriguing, do not yet pinpoint a very clear difference in developmental origin. We further agree that aberrancies in WNT signaling, seen across all age groups, suggest common biological features in GCTs of both age groups. Accordingly, we have modified the abstract to focus on the shared importance of the WNT pathway across the age groups tested in this study

3) the manuscript has potential to help in the classification of extracranial GCTs. Currently classification of testicular GCTs (WHO2016 guidelines) and ovarian GCTs (WHO 2014 guidelines) does usually not follow the same criteria. Have the authors performed correlation of the molecular data with sex? Figure 1 shows a snapshot of molecular results for the study, adding age and sex would complement the correlations and improve the results. Did the authors find differences with those cases older than 18 years old? The focus of the study were childhood and adolescents but then the results get sort of confusing in the comparisons performed.

- Response: We apologize for confusion arising from the presentation of the results. We have added annotation for age and gender to the **revised Figure 1**. We do not detect clear differences in molecular data by sex, however the study was not specifically designed to have the statistical power to address this question.

4) How were missense mutations interpreted as pathogenic? No criteria are stated in the methods. This is important to justify for example pathogenicity in APC variants. In cases in which a wnt repressor is mutated, were those mutations accompanied by a second hit? How were the wnt repressors and activators list chosen? Can the authors provide stats for the methylation clustering?

- **Response:** We have added more annotation including SIFT and Polyphen scores (revised Table S3) to clarify the predicted effect of somatic variants. We recognize that in silico predictions are not equivalent to functional studies, which are beyond the scope of this manuscript. We have added text to the presentation of these results to clarify these points.
- The list of WNT repressors and activators were chosen by reference to the list maintained by the Roel Nusse lab at Stanford (<http://web.stanford.edu/group/nusselab/cgi-bin/wnt/>) and are listed in new Supplementary Table 5. We did not perform a clustering analysis for methylation data. Figure 4A is separated by type I and II tumor types, instead of using clustering analysis.

5) Regarding WNT pathway alterations, the results are quite striking and have a lot of potential as the authors used both in vitro and in vivo models to prove a plausible effect of wnt inhibitors as a treatment. The mutations in wnt genes correlate mostly to YST subtypes, even in mixed tumors with a YST component. In intracranial GCTs genomic analysis, driver mutations in mixed tumors were the same in both histological components of the tumor, is this the case in extracranial GCT? One would expect given the results of this study that, in this case, is limited to the YST component. This could be assessed by microdissection the tissue or inferred from the tumor relative content and VAF of the mutation. Do other alterations in wnt pathway ie hypermethylation of repressors or CN gains of activators correlate to a specific subtype? Sex? Or group age that could aid in personalizing the use of these therapies?. In vitro data with wnt inhibitors shows a stronger effect in lines with a YST component. The authors used TCGA testicular dataset to validate their results but only chose 3 wnt genes, is there a reasons? Were the other wnt genes analyzed?

- **Response:** The reviewer raises interesting questions here. We deliberately focused most of our effort on samples of a single histologic subtype. Microdissection of GCTs and/or analysis of VAFs could provide insight into intratumoral heterogeneity according to histologic subtype. However our study included relatively few mixed-histology tumors, and in these cases we do not have access to the original tissue blocks to carry out microdissection. Such studies are beyond the scope of this manuscript.
- We found that alterations in WNT pathway regulators (for example hypermethylation of repressors or CN gains of activators) did not correlate to a specific histologic subtype or sex. For copy-number alterations, we were able to perform analysis on TCGA datasets to show the same patterns (copy number gain of most WNT activators and loss of most WNT repressors) in adult testicular GCT. The analysis of survival according to expression level of WNT genes was not from the TCGA data, but from an independent dataset (References 37,38). These three genes were chosen as they have been used in numerous publications as biomarkers of WNT pathway activity (see, for example, PMIDs: 20852629, 22753465, 30854133, 29029485, 24474766, 19421142, 26980182, 26403202)

6) In general, a more detailed methods section (could be included in supplements) will help in the reading and the interpretation. There are multiple examples throughout the text, i.e. the number of cases with CN rich or CN poor is not stated anywhere, and all data is given in percentages. This applies to both cases studied in this manuscript and cases from TCGA. How were gene lists chosen or why a CN of 5 was the chosen threshold is not justified, neither a range of CN observed is provided. Were multiple comparisons

considered? Fig 3 is another example> what do the % in the boxes correspond to? Number of missense mutations in each gene or number of cases mutated in a given gene?

- **Response:** We thank the reviewer for highlighting the need for more experimental detail. In Figure 4D, the number of cases with CN rich or CN poor in TCGA cohort is 85 and 71, respectively (added to text). In Figure 3, the percentage is based on the number of cases with SNVs (missense or nonsense/frameshift/splice site), or focal copy-number gains or copy-number losses, as a proportion of the total number of samples analyzed for that type of alteration. We have added this information to the figure legends.
- Regarding the threshold for copy-number changes, we now add two new supplementary figures (New Supplementary Figures 7 and 8) using a series of different CN values (3, 5, 7, 9) as thresholds to repeat the analysis in Figure 5C and 5D. As shown in Figures S7 and S8, the conclusion is always the same under all these threshold values, which confirms that our conclusions are robust and independent of threshold values. We have added a new statement to the main text to this effect.

7) the authors have germline data. Are there plausible candidate genes that could increased the risk to develop GCTs in childhood?

- **Response:** We agree that this is an important issue. We did examine our germline data, however we did not find recurrent alterations suggestive of a strong cancer predisposing effect in any one gene. This likely reflects the overall small number of germline samples included in our study, and is not too surprising given the finding of studies with much larger sample sizes that GCT lacks a set of strong susceptibility genes, for example Litchfield et al (2018) PMID: 29433971.

8) Tumor mutation burden was low in this sample set. However, the authors report several mutations in DNA repair genes. Therefore, the TMB in some of those cases (POLE, PRKDC) should be much higher (POLE is hypermutated tumors) so to drive conclusion about pathogenicity for those variants a mutational signature analysis or individual TBM analysis should be performed. Were there correlations between DNA repair mutations and cisplatin resistance?

- **Response:** In our initial analysis based on 50 tumor-normal pairs with whole-exome sequencing data, we did not find any mutation on POLE and PRKDC. However, we still include these key DNA repair genes into targeted sequencing panel on additional 81 GCTs because DNA repair genes have been proven to be important for many cancer types and were highlighted by previous publications for GCT (See, for example, PMID: 27821802, 15467433). In these additional 81 GCTs, we found 2 cases with POLE mutations and 5 cases with PRKDC mutations. However, because these results were from targeted deep sequencing with only dozens of genes, we could not accurately measure genome-wide tumor mutation burden to explore whether POLE/PRKDC mutations influence mutation burden in GCT.
- For these cases, we don't have information specifically concerning susceptibility of the tumors to cisplatin

8) Did the authors explore a link between mutations in chromatin remodelers and alteration in wnt genes methylation status?.

- **Response:** Due to the low mutation rate in GCTs and limited samples with both whole-exome sequencing and methylation data, each chromatin remodeling gene in Figure 1A only mutated in no more than one GCT case with methylation data, so we do not have statistical power to perform this analysis. We agree this will be an interesting possibility to pursue in future studies.

9) It is striking the number of mutations found in FAT genes. FAT genes functions are still unclear although FAT1 is known to activate wnt pathway and other roles in development have been postulated. Is

CTNNB1 highly expressed and nuclear in FAT2, 3 mutated cases? FAT4 is the closest to the drosophila fat gene and therefore might be implicated in hippo pathway. Were other hippo genes mutated in type I or II GCTs. The authors say that FAT 4 is mutated in type II tumors but the Figure 1A does not show that correlation, could this be a mistake?

- **Response:** To explore whether CTNNB1 is highly expressed in FAT2/FAT3 mutated cases, we now add a new supplementary figure (**new Supplementary Figure 1**) by integrating RNAseq and whole-exome sequencing data. As shown in Figure S1, we found a significantly increased CTNNB1 expression in GCT cases with somatic protein-altering mutations in FAT2 or FAT3 genes compared to GCT cases without such mutations (Mann-Whitney U test, $P=0.001$).
- We used all hippo signaling pathway genes defined by KEGG database (https://www.genome.jp/kegg-bin/show_pathway?hsa04390), however, none of them were found to have somatic mutations in our cohort.
- We thank the reviewer for pointing out this discrepancy. All four cases with FAT4 mutations are females with ages 1, 9, 15, and 16. The last three with age 9, 15, and 16 are ovarian GCTs that lack 12p amplification and therefore were designated Type I. We have corrected the text.

Reviewer #2 (Remarks to the Author): Expert in computational biology

Several molecular studies suggested distinct molecular origins and developmental mechanisms of type I and type II Germ cell tumors (GCTs). In this study authors established the activation of the WNT pathway as a molecular signature of both type I and type II GCTs. Authors have investigated several genomic aspects of GCTs—somatic mutations, copy-number alteration, and differential promoter methylation—in relation WNT pathway. In addition, they have also shown that WNT inhibitors can suppress GCT cells both in vitro and in vivo suggesting novel targeted therapies. In general, the design of experiments and data analysis were well-planned, covering most of the key aspects in integrative genomic studies. Addressing the following comments below would further strengthen and better comprehend the scientific advances.

1. What fraction of the whole-exome sequencing and custom-capture deep sequencing overlap with RNA-Seq data? Authors reported stop-gain mutations in APC and FAT1 that likely result in truncated proteins. Do authors find any truncated transcripts in the RNA-Seq data?

- **Response:** As indicated in Table S1, among 131 GCT cases with either the whole-exome sequencing and custom-capture deep sequencing data, 26 of them have RNAseq data. Unfortunately, the case with APC and FAT1 stop-gain mutations did not have available RNA for conducting RNA sequencing.

2. Did authors perform any base quality recalibration prior to variant calling? Authors used a read quality filter of ≥ 15 . However, score of 20 (99%) or 30 (99.9%) are typically recommended.

- **Response:** We agree with the reviewer. We have performed base quality recalibration prior to variant calling, and used read quality score > 30 in variant calling. We have added this information to the methods.

3. A significant fraction of the analysis is based on the WNT CN-rich and WNT CN-poor classification. What is the rationale of using “5” as a hard threshold? Did authors try other cutoffs? If so, how did they choose “5”? A note of this in the methods section may benefit other researches performing similar analyses.

- **Response:** Please see response to reviewer 1 above. We now add two new supplementary figures (**new Supplementary Figures 7 and 8**) by using a series of random CN values (3, 5, 7, 9) as thresholds to repeat the analysis in Figure 5C and 5D. As shown in Figures S7 and S8, the

conclusion is always the same under all these threshold values, which confirms that our conclusions are robust and independent of threshold values. We have clarified this point in the text.

4. Authors proposed/reported activation of the WNT pathway is associated with poor clinical outcomes in both children and adolescents. However, the analysis is not performed independently for type I and type II.

- **Response:** Owing to the overall small number of relapse cases, our study was not adequately powered to perform the comparison between Type I and Type II for this question. This will be the subject of future studies.

Minor points:

1. Introduction: novels therapies -> novel therapies
2. Results: Consider rephrasing “The 51 genes were ...”
3. Methods: Raw reads was mapped -> Raw reads were mapped

- **Response:** Thank you for pointing out these issues. The text has been revised accordingly

Reviewer #3 (Remarks to the Author): Expert in pediatric oncology

The manuscript “Integrated genomic analysis reveals aberrations in WNT signalling in germ cell tumors of childhood and adolescence” presents a range of sequencing analyses from a cohort of germ cell tumours and presents evidence to suggest that in at least some of these, WNT signalling activation is the driver mechanism.

The major strengths of the manuscript are the presentation of the genomic features of a tumour type that has, in paediatric cases at least, been relatively understudied. The manuscript also presents a functional analysis of the contention that WNT activation is a driver in a proportion of these tumours.

The weaknesses are that the more comprehensive sequencing approaches (WES and RNA seq) are applied only to a small number of the cohort, and the majority of samples do not have exome sequencing or RNA sequencing. This will necessarily restrict the potential to identify the full range of variants that might be observable in the tumours. In addition, beyond the assessment of copy number variants, there appears not to be any other analysis for other structural variants, such as fusions, which might be available from RNA data. Thus whilst I am enthusiastic about a manuscript addressing the genomic variants in less-studied tumour types, my concern is that the data presented in this paper is insufficiently comprehensive and generalizable.

Listed below are more specific questions and comments for the authors.

1. Whilst I appreciate the significant work involved in this suggestion, if there were the opportunity to expand the WES and RNAseq in the cohort, that would be highly desirable.

- **Response:** We agree that ideally there would be more cases with full omics analysis. However, many samples for which we performed RNA-Seq are not eligible for whole-exome sequencing owing to the specific informed consent guidelines under which those samples were obtained. In the future, we hope to readdress this approach with more samples collected using genomics-specific informed consent documents, such as the Children’s Oncology Group Project Every Child which recently began enrolling patients.

- To address the point raised here about structural variants, we have performed a new whole-genome sequencing analysis in 10 tumor-normal pairs at 30x resolution. We used the DEFOR (PMID: 30860569) and SCHALE (PMID: 23613679) algorithms to assess copy-number changes, structure alterations and loss of heterozygosity (LOH) in the tumors. The results are shown in new Supplementary Figures 3-5. We observed somatic (tumor-specific) recurrent focal- and arm-level structural alterations events, recapitulating those described by lower-resolution array technologies in our study and by other groups, such as 1p gain, 6q loss and 12p gain. Of note, an ovarian pure yolk sac tumor from a 23 year-old female did not exhibit any evidence of chromosome 12p gain, supporting the idea that such tumors are more closely related to Type I YSTs of young children. We also observed novel copy-number changes and loss of heterozygosity (LOH) events. Further using the DEFUSE algorithm on our RNASeq data, we report computationally-identified gene fusions that have not been reported previously (new Supplementary Table 4). The functional significance, if any of these genetic alterations will be explored in our follow-up analysis. In addition, the data have been made available along with our other results and may be useful for other investigators.

2. With the variants described in the WNT genes in Figure 1 and the supplemental information, what evidence supports these mutations as pathogenic or driver mutations? Indeed, this same question could be addressed for the non-synonymous and splice variants (other than established hot-spot mutations) identified in the cohort. The point here being to distinguish, in some way, true driver mutations from across the cohort.

- Response: Please see response to reviewer 1 above. We have added more annotation including SIFT and Polyphen scores (revised Supplementary Table 3) to clarify the predicted effect of somatic variants. We recognize that in silico predictions are not equivalent to functional studies, which are beyond the scope of this manuscript. We have added text to the presentation of this results to clarify these points.

3. In Figure 2, why was the age cut-off selected to distinguish type I from type II GCTs when the histology is available? In table 1 there seem to be a number of YST in patients >6 yrs, which suggests that age segregation might not be completely accurate. Further, what distinct molecular origins are the authors proposing that give rise to type I and type II GCT? Are there other data they have which might support these origins? Is the suggestion that the significant association of whole chromosome LOH with the <6yo group indicates an origin of a mis-segregation event followed by chromosome duplication? Is this supported by the karyotype data? Are there mutations in these tumours that are associated with this mechanism? Are the LOH copy neutral or associated with copy number gain or loss?

- Response: Several important points are raised by the reviewer. We agree that age segregation is not a foolproof way to distinguish Type I from Type II tumors. Though epidemiologic data largely support a distinction between childhood vs. adolescent/adult GCTs, which is further reinforced by the limited histologic appearance of Type I tumors (teratomas and YSTs only), there are important exceptions. Dysgerminomas or mixed malignant ovarian GCTs (by definition Type II) can occur in girls less than 11 years of age. On the other hand, some pure ovarian YSTs lack evidence of 12p amplification and thus are more properly designated Type I, as we have done in our analysis. We have added the age and gender data to Figure 1A, which clarifies how individual tumors were classified in our study.
- The observations made in our study provide further support for molecular, and not just histologic, distinctions between Type I and Type II tumors. These include the lower incidence of arm-level LOH, and the higher levels of WNT signaling activity, associated with Type I tumors. We agree that karyotype data might provide some insight into chromosome segregation aberrancies in GCTs, however karyotype data are not available for the tumors we analyzed. As our data do not

speak specifically to the molecular mechanisms underlying these differences, we have removed the text speculating on distinct molecular origins from the manuscript.

- For the question “Are the LOH copy neutral or associated with copy number gain or loss?”, LOH is due to a change from a heterozygous state in the germline to an apparently homozygous state in the tumour DNA. Therefore, LOH encompasses both LOH with copy number losses (termed as CNL-LOH) and copy number neutral LOH (termed as CNN-LOH). For CNL-LOH, all or part of a chromosome is deleted and therefore largely overlaps with copy-number loss events, while CNN-LOH still maintains original copy numbers and therefore should not be associated with either copy-number gain or loss events. In our analysis, we include both CNL-LOH and CNN-LOH, as defined by most published studies to study LOH patterns in human cancer.

4. I could not see, in the main body of the manuscript, that the authors distinguished focal copy number losses as heterozygous or homozygous. Similarly, I could not determine if copy number gains were quantitated in any way.

- Response: We based our analytic approach on that used by The Cancer Genome Atlas (TCGA) project, which has developed well-accepted criteria to analyze copy-number status in human tumor genomic data, as shown in the TCGA copy-number pan-cancer analysis paper (PMID: 24071852) and TCGA/GDC web portal (https://docs.gdc.cancer.gov/Data/Bioinformatics_Pipelines/CNV_Pipeline/), which states that “... their numeric CNV values were further thresholded by a noise cutoff of 0.3: Genes with focal CNV values smaller than -0.3 are categorized as a "loss" (-1), and Genes with focal CNV values larger than 0.3 are categorized as a "gain" (+1)”. We applied these criteria to define copy-number gain or loss in our analysis in order to be consistent with TCGA. The TCGA/GDC web portal does not define heterozygous or homozygous copy number loss status, so in our current manuscript we also have not done that. But we agree this would be interesting to pursue in our follow-up studies. Figure 2A presents GISTIC analysis for assigning statistical significance to copy-number gains and losses.

5. Did the authors look for other types of structural variants?

- Please see our response to Point 1 above. We report predicted gene fusions (new Supplementary Table 4).

6. The WNT observation is, I agree, intriguing. The gene expression data was confusing as it was not clear whether expression data derived from this cohort (32 RNAseq samples) was used to define beta-catenin expression. Was this RNA data used to look more broadly at gene expression? I did not understand why a cut-off of 5 or more copy number alterations was chosen. At first glance, this seemed quite arbitrary. The relationship between 5 or more CNV and beta catenin expression also seems a little unclear. Why would 5 or more copy number changes be required to elevate beta-catenin expression? What is the effect of each of the copy number changes on the expression of that individual gene? Are these heterozygous or homozygous copy number losses? Is there evidence to support haploinsufficiency of these genes as a mechanism of WNT pathway activation?

- Response: Expression data derived from this cohort (32 RNAseq samples) were used to define beta-catenin expression. Figure 5B is based on RNAseq data derived from this cohort (32 RNAseq samples), while the panels in Figure 5E are based on an independent large GCT cohort with survival data.
- Regarding copy-number threshold, please see response to reviewer 1 above. We now add two new supplementary figures (new Supplementary Figures 7 and 8) by using a series of random CN values (3, 5, 7, 9) as thresholds to repeat the analysis in Figure 5C and 5D. As shown in Figures

S7 and S8, the conclusion is always the same under all these threshold values, which confirms that our conclusions are robust and independent of threshold values. We will add a new statement in the main text.

- We appreciate that this reviewer brings an important question for the relation between copy number changes and gene expression. We recently published a new Bayesian statistics algorithm, called iExCN (PMID: 29972784), to provide a comprehensive analysis on how each of the copy number changes will influence the expression of that individual gene. We plan to implement this algorithm into analyzing this GCT cohort in our next follow-up study. Because the scope and complexity of this iExCN analysis, we will not include it into this manuscript.
- For the question on “heterozygous or homozygous copy number losses”, if we define zero copy of DNA as “homozygous” copy number losses, we found that no GCT patient actually have zero copy of WNT genes. In other words, all copy number loss events studied here are “heterozygous copy number losses”
- We do not have evidence either way to support haploinsufficiency of these genes as a mechanism of WNT pathway activation. Assessing this possibility will require extensive functional genomic experiments which we feel are beyond the scope of the current manuscript

7. In reference to data in Figure 5, the authors state “To determine the possible prognostic significance of WNT pathway activation, we tested the association between WNT gene copy-number alterations (gain of WNT activators and loss of WNT repressors) and outcome in our dataset”. Are these the same CNV that are described in Figure 4? **YES**

- Response: Yes, these are the same copy number alterations. We have added new Supplementary Table 5 to provide more clarification on these genes

8. Is the *PIWIL1* gene subject to CNV in GCT? Why isn't this gene included in the data in Figure 4 or 5?

- Response: *PIWIL1* experienced 19 gains and 8 losses in 148 GCT samples in total. The frequency is relatively low so CNV might not be the major driving force.

Reviewer #4 (Remarks to the Author): Expert in zebrafish cancer models

Key Results:

In this manuscript the authors use zebrafish to determine the role of Wnt signaling in GCT progression. They utilized a previously established zebrafish model of GCT which has a loss-of-function mutation in *bmpr1bb* leading to impaired spermatogonial differentiation and testicular tumours.

The authors used adult male *bmpr1bb*; *piwil1:EGFP* fish in which GCT tumours had developed which were then treated with 1 μ M of IWR-1 for 7 days. In the treated fish they observed a loss in EGFP fluorescence and an increase in γ H2AX. These results suggest that inhibition of Wnt signaling is causing differentiation of the GCT cells as γ H2AX is a marker of meiosis.

Originality, significance, validity and conclusions:

The zebrafish results are an elegant addition to the manuscript, providing in vivo data to support the bioinformatic, epidemiological and in vitro data.

Statistics:

(Figure 6 only)

Overall the statistics used are appropriate. However, the number of fish and any experimental replicates should be included in the legend.

- Response: This information has been added to the figure legend.

Referencing:

Appropriate.

Clarity and context

Well written.

Suggested Improvements

1) The authors propose that the GCT tumours treated with IWR-1 are more ‘differentiated’ but this is based only on the pH2AX staining. Given that this is a key outcome from the study it would be beneficial to have further evidence to support this claim.

Perhaps H&E staining of the control vs IWR-1 treated gonads could reveal histological changes supporting this claim?

- Response: We thank the reviewer for this suggestion. Indeed, histologic examination of H&E-stained tumor sections showed that control DMSO-treated tumors consisted of sheets of spermatogonia and some primary spermatocytes, with only scattered islands of mature spermatozoa, as we previously described (PMID 20047465). In contrast, tumors from IWR-1 treated fish exhibited markedly more complete differentiation, with many lobules showing the full range of spermatocytic differentiation (new Figure 6F).

2) It is not clear from the manuscript what the significance is for the loss of piwil1:EGFP expression in IWR-1 treated *bmpr1bb*^{-/-} mutants vs no change in WT animals (Fig 6F). Presumably as piwil1 is Wnt-regulated and upregulated in GCT this is a control for the efficacy of IWR treatment? Initially I thought piwil1:EGFP expression was used as a readout of tumour size and/or differentiation status, but I am not sure if this was the authors intent. This should be clarified.

- Response: We apologize that this section was not clearly written. We have revised the text and the figure to be more clear: “We previously reported that male zebrafish bearing mutations in the *bmpr1bb* gene develop testicular GCTs with elevated piwil1 expression¹⁹. Using a piwil1:eGFP transgenic zebrafish reporter line⁴⁷ that permits live visualization of GCTs in *bmpr1bb* mutants (Supplementary Figure 9), we tested the effects of WNT inhibition. We treated males with GCTs for 7 days of with DMSO vehicle control or with IWR-1 (Figure 6C,D). Treatment with IWR-1 led to a striking decrease of eGFP expression, with the eGFP signal in treated fish reverting to the levels seen in non-tumor-bearing wildtype fish (Figure 6E), indicating that WNT inhibition downregulates activity of the piwil1 promoter.”

3) Panels 6C and 6D should be labeled to indicate that the fish were bmpr1bb-/-, it would also be valuable to include the piwill:EGFP expression in WT fish for comparison.

- Response: Figures 6C and 6D have been revised. New Supplementary Figure 9 includes WT fish for comparison.

** See Nature Research's author and referees' website at www.nature.com/authors for information about policies, services and author benefits.

COVID 19 and impact on peer review

As a result of the significant disruption that is being caused by the COVID-19 pandemic we are very aware that many researchers will have difficulty in meeting the timelines associated with our peer review process during normal times. Please do let us know if you need additional time. Our systems will continue to remind you of the original timelines but we intend to be highly flexible at this time.

This email has been sent through the Springer Nature Tracking System NY-610A-NPG&MTS

Confidentiality Statement:

This e-mail is confidential and subject to copyright. Any unauthorised use or disclosure of its contents is prohibited. If you have received this email in error please notify our Manuscript Tracking System Helpdesk team at <http://platformsupport.nature.com>.

Details of the confidentiality and pre-publicity policy may be found here <http://www.nature.com/authors/policies/confidentiality.html>

Privacy Policy | Update Profile

DISCLAIMER: This e-mail is confidential and should not be used by anyone who is not the original intended recipient. If you have received this e-mail in error please inform the sender and delete it from your mailbox or any other storage mechanism. Springer Nature Limited does not accept liability for any statements made which are clearly the sender's own and not expressly made on behalf of Springer Nature Ltd or one of their agents.

Please note that Springer Nature Limited and their agents and affiliates do not accept any responsibility for viruses or malware that may be contained in this e-mail or its attachments and it is your responsibility to scan the e-mail and attachments (if any).

Springer Nature Ltd. Registered office: The Campus, 4 Crinan Street, London, N1 9XW. Registered Number: 00785998 England.

REVIEWERS' COMMENTS

Reviewer #1 (Remarks to the Author):

The manuscript has been significantly improved. I have one outstanding point. Higher expression levels of CTNNB1 are shown but does it accumulate in the nucleus? one would simply hypothesize that yes. However, given the robustness of CTNNB1 IHC as a surrogate of Wnt activation why not use this as a marker for selection of wnt associated cases? I would strongly suggest to the authors to stain the samples for the CTNNB1 marker, this could provide a fast tool to select patients for treatment for example (given than the authors hypothesize about this possibility).

Reviewer #2 (Remarks to the Author):

All concerns have been addressed.

Reviewer #3 (Remarks to the Author):

I appreciate the efforts that the authors have made in addressing my questions, and in modifying their manuscript in light of these (and those of the other reviewers). The general strengths of the manuscript remain as first indicated - it is presented some interesting genomic data on a relatively understudied tumour population, and is backing up some of the observations with direct experiments in model systems. The authors have also addressed many/most of my comments and added significant new data. The weakness that remains is the relatively small cohort that is under study, and the relatively small number of samples with comprehensive genomic analyses. However, in the revised paper, I think the strengths outweigh the weaknesses. I have no further requests for changes or additional comments to make.

Reviewer #4 (Remarks to the Author):

The authors have addressed my concerns. I only have one minor comment:

1) Please add scale bars to Figure 6C, 6D and 6F.

REVIEWERS' COMMENTS

Reviewer #1 (Remarks to the Author):

The manuscript has been significantly improved. I have one outstanding point. Higher expression levels of CTNNB1 are shown but does it accumulates in the nucleus? one would simply hypothesise that yes. However, given the robustness of CTNNB1 IHC as a surrogate of Wnt activation why not use this as an marker for selection of wnt associated cases? I would strongly suggest to the authors to stain the samples for the CTNNB1 marker, this could provide a fast tool to select patients for treatment for example (given than the authors hypothesise about this possibility).

Response: We agree with the reviewer that CTNNB1 could be a valuable IHC marker for germ cell tumors. It certainly makes sense to look at beta-catenin via IHC, given what we think is the importance of WNT signaling. There are some challenges. Assembling a cohort of tumors for IHC will pose some difficulties. Germ cell tumors are rare; for the primary genomic study it took us several years to collect the tumors, and in the vast majority of cases we received purified DNA or RNA from contributing sites. For this reason we have very few banked tumor blocks. However, going forward we will attempt to identify such tissue for the correlative studies.

The reviewer's idea of using beta-catenin IHC as a risk stratification biomarker is a very good one. Because adverse outcomes are relatively rare in this disease, it would require a large sample size to validate such a marker. In fact, we have this idea built into the current Children's Oncology Group clinical trial (AGCT1531), where tumor tissue is being collected for precisely such assays. However, that trial will still be continuing for several years before samples are available. For these reasons we think the requested studies might be better suited for a follow-up manuscript

Reviewer #2 (Remarks to the Author):

All concerns have been addressed.

Response: Thank you for reviewing the initial and revised manuscripts, and for your helpful comments.

Reviewer #3 (Remarks to the Author):

I appreciate the efforts that the authors have made in addressing my questions, and in modifying their manuscript in light of these (and those of the other reviewers). The general strengths of the manuscript remain as first indicated - it is presented some interesting

genomic data on a relatively understudied tumour population, and is backing up some of the observations with direct experiments in model systems. The authors have also addressed many/most of my comments and added significant new data. The weakness that remains is the relatively small cohort that is under study, and the relatively small number of samples with comprehensive genomic analyses. However, in the revised paper, I think the strengths outweigh the weaknesses. I have no further requests for changes or additional comments to make.

Response: Thank you for reviewing the initial and revised manuscripts, and for your helpful comments.

Reviewer #4 (Remarks to the Author):

The authors have addressed my concerns. I only have one minor comment:

1) Please add scale bars to Figure 6C, 6D and 6F.

Response: Scale bars have been added as requested